# Exciton-Polariton Condensates: A Fourier Neural Operator Approach

**Surya T. Sathujoda** [*]
University of Cambridge
sts40@cam.ac.uk

**Yuan Wang** [*†]
University of Southampton
yw2e17@soton.ac.uk

**Kanishk Gandhi**
University of Southampton
kg7g19@soton.ac.uk

## Abstract

Advancements in semiconductor fabrication over the past decade have catalyzed extensive research into all-optical devices driven by exciton-polariton condensates. Preliminary validations of such devices, including transistors, have shown encouraging results even under ambient conditions. A significant challenge still remains for large scale application however: the lack of a robust solver that can be used to simulate complex nonlinear systems which require an extended period of time to stabilize. Addressing this need, we propose the application of a machine-learning-based Fourier Neural Operator approach to find the solution to the Gross-Pitaevskii equations coupled with extra exciton rate equations. This work marks the first direct application of Neural Operators to an exciton-polariton condensate system. Our findings show that the proposed method can predict final-state solutions to a high degree of accuracy almost 1000 times faster than CUDA-based GPU solvers. Moreover, this paves the way for potential all-optical chip design workflows by integrating experimental data.

## 1 Introduction

The rapid advancement of exciton-polariton condensates [1] has led to the emergence of a wide range of all-optical devices, such as switches [2–8], analogue simulators [9, 10], neuromorphic computing [11–15], transistors [16–18], etc. The microcavity exciton-polariton (hereafter polariton) [19] system consists of two main strongly coupled components: quantum well (QW) excitons and photons trapped in microcavity. The former are electron-hole pairs bound by Coulomb interactions, are frequently observed in semiconductor QWs, and the latter are generated from the distributed Bragg reflectors (DBRs) aiming to create a stopband in the refreactive spectrum of the microcavity structure [19], as illustrated in Figure 1. In this system, the exciton lifetime significantly surpasses that of photons in microcavity. Thus, these high-quality DBRs, engineered to extend the photon lifetime, ensure the preservation of the strong-coupling condition, where the light-matter coupling strength surpasses the decay rate of any component within the polaritonic system [20].

Polaritons, often described as quasiparticles with characteristics that are half-light and half-matter, have an impressively low effective mass due to their photonic components. This mass is approximately five orders of magnitude less than that of a bare electron. Thus, the necessary temperature for condensation is approximately $10\,\mathrm{K}$ for inorganic semiconductor materials [1], which contrasts

---

[*]Equal contribution.
[†]Corresponding author.

NeurIPS 2023 AI for Science Workshop.

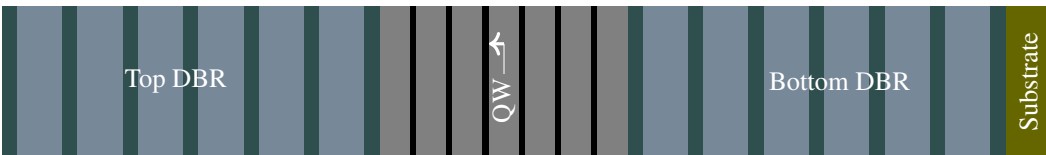

Figure 1: Sketch of the distributed Bragg reflectors (DBRs) microcavity integrated with quantum wells (QWs). The DBR is comprised of 6 bilayers of alternating low (depicted in light blue) and high (shown in dark green) refractive index materials. Also, 6 QWs (represented in black) with their respective barriers (in gray) are depicted. The substrate, located at the base of the structure, is presented in dark yellow.

significantly with atomic condensates such as Rubidium-87, which require temperatures around $170\,\text{nK}$ [21]. Notably, using organic materials, polariton condensation can be realized even at room temperature [22–24]. In polariton condensates, due to their excitonic component, the predominant repulsive interaction between polaritons results in a blue-shifting effect on the potential and gives rise to rich nonlinear effects [25, 26]. In essence, it is the photonic (light-like) component of the polaritons that confers their notably light effective mass suitable for condensates, while the excitonic (matter-like) component is responsible for the observed nonlinear effects.

The emergence and adoption of all-optical devices necessitates the development of precise and adaptable simulation tools. Just as Electronic Design Automation (EDA) played a pivotal role in the evolution of chip design, there is a pressing need for emulators that can capture the rich of nonlinear characteristics inherent in optical devices. A case in point is the groundbreaking development of an optically-activated transistor switch. Rooted in advancements in polariton condensates, this transistor was initially conceptualized for operations at cryogenic temperatures [16]. However, strides in research have broadened its applicability, making it feasible even under ambient conditions [17, 18]. These rapid advancements emphasize the need for comprehensive simulations in the design of large-scale logic gates. Hence, crafting a robust theoretical framework becomes imperative, one that can effectively allow manipulation of the potential profile via specific input configurations.

To this aim, a robust solution to the Gross-Pitaevskii equation (GPE) which is used to theoretically describe the condensates is required [27]. While GPU-based GPE solvers tailored for both uniform [28–31] and non-uniform meshes [32] currently exist, we aim to adopt an even more efficient machine-learning (ML) based solver that is intended to notably accelerate the computational process, especially in the context of designing extensive transistor networks.

Thus far, a wide variety of ML architectures have been proposed to approximate solutions to general classes of PDEs. These range from convolution-based methods, such as variants of the U-Net architecture [33], to operator-learning methods such as Deep Operator Networks (DeepONets [34]), Graph Neural Operators [35], Multipole Graph Neural Operators [36], Fourier Neural Operators [37] and Physics-informed Neural Operators [38]. Though convolution-based methods have shown promise regarding accuracy of future state predictions, they fail to scale in compute efficiency to larger systems, even with recent advancements [39]. Operator-learning methods overcome this bottleneck by learning mappings between infinite-dimensional spaces, allowing them to predict solutions at different discretiations at a similar speed.

In this work, we study the application of the Fourier Neural Operator (FNO) architecture to approximate solutions to the GPE. We are interested in this specific variant of Neural Operator as its mathematical formulation relates very closely to the Split-step Fourier Method (SSFM), which is the numerical method used to solve the GPE in this instance. Moreover, FNOs have shown widespread success in application to many other areas of physics and engineering [40–43].

The structure of this paper is outlined as follows: In Section 2, we provide an introduction to exciton-polariton condensates and explain the physical quantities employed within the GPE. Section 3 introduces the Neural Operator architecture, detailing its compatibility with the GPE, especially when integrated with the rate equation. This section further elaborates on data generation and preparation methods. Our findings are presented in Section 4, future work in Section 5 and the general conclusions are given in Section 6.

## 2  Background

### 2.1  Exciton-polariton condensates

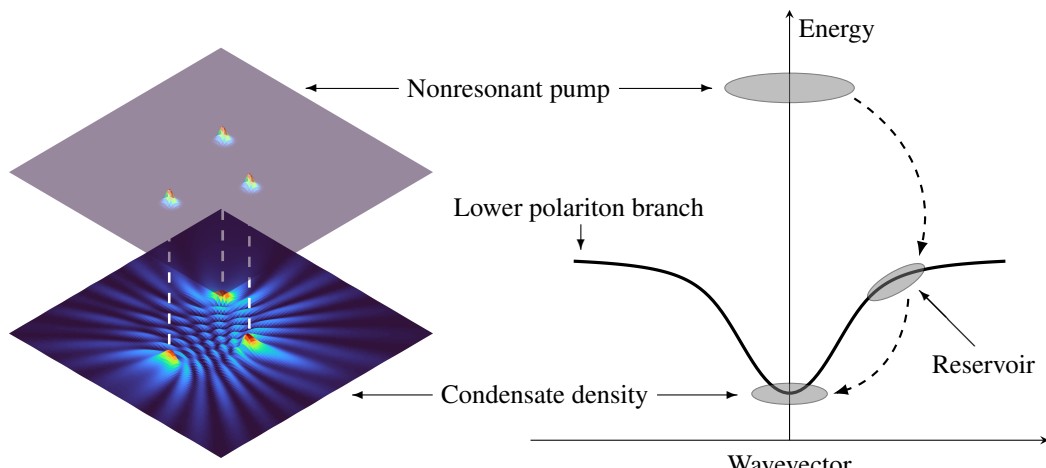

Figure 2: Comparison of pump profiles and wavefunction density with scattering process illustration. Left column: The upper panel shows the nonresonant pump profile featuring three Gaussian spots, while the lower panel shows the wavefunction density of the condensates at the final time. Three white dashed lines indicate the central positions of the pump regions and align with their corresponding locations on the condensate density map. Right column: Depiction of the scattering process, tracing the transition from the hot electron-hole plasma phase, through the reservoir cooling phase, to the scattering in the condensates. Only the lower polariton branch of the polariton energy mode is shown here.

Polariton condensates, being an non-equilibrium process, start with nonresonant excitation (see the right column of Figure 2). The hot electron-hole plasma initially undergo rapid cooling, primarily driven by emissions mediated by the longitudinal-optical phonons, forming excitions at the high momentum of the lower polariton (LP) branch. Subsequent interactions between excitons and acoustic phonons, as well as exciton-exciton scatterings, lead polaritons to a transitional 'bottleneck region' within the LP branch, as illustrated in Figure 2. It is worth noting that the terminology 'polariton' instead of 'exciton' is used here because at very high position of the LP branch the photonic components of the polaritons are barly present and then the quasiparticles arrive all the way to the bottleneck region where more photons are coupled. By the end of the process, parametric scattering, together with momentum conservation, takes place, which results in a segment of the polaritons to transition to a higher-momentum state, while another segment descends into the lower-momentum branch of the LP forming condensates. Ballistic polariton flow (see the left column of the Figure 2) is activated once the system is above the condensate threshold [44]. These condensates represent large-scale coherent states, distinct from the initial nonresonant inputs [1].

Technological advances, such as the use of spatial light modulators and precise semiconductor fabrications, allow the manipulation of the pump profile and rich nonlinear condensate outputs, leading to engineering of quantum fluids of light [45]. Extensive studies have been conducted on the ability of nonresonant optical techniques to manipulate the motion of condensate polaritons, such as customized momentum distributions [46], condensate amplifiers [44, 47], waveguides [48–51], directional superfluids near equilibrium [52]. The ability to manipulate the condensate flow through reservoir engineering shows its potential for quantum computing [53].

## 2.2 Gross-Pitaevskii equation

The dynamics of polariton condensates can be described by the GPE coupled with the rate equation of the exciton reservoir denoted as $\mathcal{N}$ [27]:

$$i\hbar\frac{\partial}{\partial t}\Psi = \left\{ -\frac{\hbar^2}{2m}\nabla^2 + \alpha|\Psi|^2 + G\Big[\mathcal{N} + \frac{\eta}{\Gamma}P(\boldsymbol{r})\Big] + i\frac{\hbar}{2}\big[R\mathcal{N} - \gamma\big] \right\}\Psi, \tag{1}$$

$$\frac{\partial}{\partial t}\mathcal{N} = -\Big[\Gamma + R|\Psi|^2\Big]\mathcal{N} + P(\boldsymbol{r}), \tag{2}$$

where $m$ is the effective mass of the polariton, $\alpha$ and $G$ stands for, respectively, polariton-polariton and polariton-reservoir interaction, $R$ denotes the scattering rate from the reservoir to the condensates, $\eta$ refers to the ratio of the dark excitons, and $\gamma$ ($\Gamma$) is the decay rate of the polariton (reservoir). The detunning between the exciton and the photon mode can greatly alter the interaction terms with the relationship $\alpha = g|\chi|^4$ and $G = 2g|\chi|^2$, and $g = g_0/N$ where $g_0$ is the exciton-exciton interaction, $N_{\text{QW}}$ the number of QWs, and $|\chi|^2$, representing the presentage of exciton of which the polariton consists, is the Hopfield coefficient [54] of the excitonic branch. In this work, the continuous-wave (CW) pump, denoted by $P(\boldsymbol{r})$, is used to replenish the reservoir due to the losses from the system. The nonlinear term $|\psi|^2$ appearing in both the pump-to-reservoir transition [see Equation (2)] and the superfluid in the condensates [see Equation (1)], produce the rich nonlinear characteristic induced from the pump to the condensate.

In the case of CW excitation under a weak pumping regime, the approximate value of $|\psi|^2$ tends towards zero. In this situation, the rate of reservoir with respect to time maintains a steady state, or in mathematical terms, $\partial\mathcal{N}/\partial t = 0$. The determination of threshold power, denoted at $p^{\text{th}}$, is possible through an analysis of the right-hand side (r.h.s.) of Equation (1) where $R\mathcal{N} = \gamma$ serves as a representative of the equilibrium state between gain and loss. Therefore, the threshold power $p^{\text{th}} = \gamma\Gamma/R$ is obtained. This suggests that when the population of polaritons exceeds the condensation threshold $p^{\text{th}}$, a detectable density value manifests itself. The real potential of Equation (1) denoted $V$ in the stationary state of the system, therefore, is

$$V(\boldsymbol{r}) = \alpha|\psi|^2 + G\Big(\frac{1}{\Gamma + R|\psi|^2} + \frac{\eta}{\Gamma}\Big)P(\boldsymbol{r}). \tag{3}$$

The real potential is composed of two main components: one originating from the pumping region [first term on the r.h.s of Equation (3)] and the other stemming from the interactions among the polaritons outside this region [second term on the r.h.s of Equation (3)]. When the pumping power is below the threshold, the direct contribution of the potential goes directly into the pumping profile. This relationship is represented as $V(\boldsymbol{r}) = (1 + \eta)(G/\Gamma)P(\boldsymbol{r})$. The spatial profile chosen for the demonstration of $N_G$ Gaussian spots. That is

$$P(\boldsymbol{r}) = \sum_i^{N_G} p_i G_i(\boldsymbol{r}), \tag{4}$$

where $p_i$ stands for strength of each spot and the normalized Gaussian function $G_i(\boldsymbol{r})$, with full width at half maximum (FWHM) denoted $\sigma$, is defined as

$$G_i(\boldsymbol{r}) = \frac{1}{2\pi\sigma}\exp\left(\frac{-|\boldsymbol{r} - \boldsymbol{r}_i|}{2\sigma^2}\right). \tag{5}$$

Note that $\boldsymbol{r}_i$ represents different location of spots.

# 3 Methodology

## 3.1 Fourier Neural Operators

The numerical solution to Equations (1) and (2) is derived using the SSFM, detailed in Appendix A. A natural ML analog to this classical method is the FNO architecture [37]. More generally, Neural Operators [55] are a class of models which learn mappings between two infinite-dimensional spaces from a finite set of input-output pairs. Many variants of the Neural Operator architecture have been applied to approximate solutions to Partial Differential Equations, such as in [40–43]. The Neural

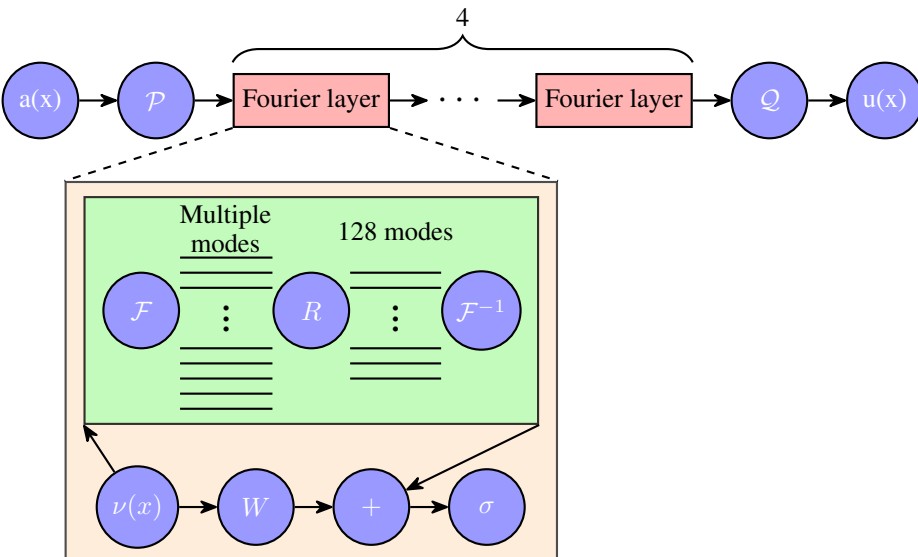

Figure 3: Architecture of the Fourier Neural Operator: The process begins with the input $a(x)$ which undergoes a lifting operation, denoted as $\mathcal{P}$. This is followed by 4 consecutive Fourier layers. Subsequently, a projector $\mathcal{Q}$ transforms the data to the desired target dimension, resulting in the output $u(x)$. The inset provides a detailed view of the structure of a Fourier layer. Data initially flows to the layer as $\nu(x)$ and is bifurcated into two branches: one undergoes a linear transformation $W$, and the other first experiences a Fourier transformation, from which the 128 lowest Fourier modes are kept and the other higher modes are filtered out by undergoing a transformation $R$, and ends with an inverse Fourier transformation with these left modes. The two data streams then converge, followed by the application of an activation function $\sigma$.

Operator architecture consists of a lifting operation $\mathcal{P}$, followed by iterative updates using a Kernel Integral Operator $\mathcal{K}$, and a final projection operator $\mathcal{Q}$, as defined in Equation 6.

$$\mathcal{G}_\theta := \mathcal{Q} \circ \sigma_T(W_{T-1} + \mathcal{K}_{T-1} + b_{T-1}) \circ ... \circ \sigma_1(W_0 + \mathcal{K}_0 + b_0) \circ \mathcal{P} \tag{6}$$

Here $\sigma$ corresponds to a non-linearity and $W$ and $b$ correspond to the weights and biases of the Kernel Integral Layer, respectively. $\mathcal{P}$ and $\mathcal{Q}$ are point-wise fully local projection and lifting operators. The choice of the Kernel Integral Operator $\mathcal{K}$ delineates the class of the Neural Operator. Specifically, the FNO (see Figure 3) uses the Kernel Integral Operator defined by Equation 7 below.

$$(\mathcal{K}_t(v_t))(x) = \mathcal{F}^{-1}(R_\phi \cdot \mathcal{F}(v_{t-1}))(x) \qquad \forall x \in \mathbb{R}^n \tag{7}$$

Here $\mathcal{F}$ and $\mathcal{F}^{-1}$ correspond to the Fourier and Inverse Fourier Transforms and $R_\phi$ corresponds to the Fourier Transform of a periodic function arising from the definition of a Kernel Integral Operator given in [37]. This object is parameterised by a linear transformation of the top $k$ modes pertaining to the given layer, which acts as a hyperparameter in the model.

The natural choice of the FNO architecture for approximating the solution to Equations 1 and 2 is due to the inductive bias that arises from the SSFM - FNO correspondence stated below.

**Theorem 1.** *(SSFM-FNO Correspondence) Suppose that $\sigma \in (TW)$ is a Tauber-Wiener function, $X$ is a Banach Space, $K \subset X$ is a compact set, $V$ is a compact set in $C(K)$, $\Psi_t$ is a nonlinear continuous operator representing the solution of the first-order Split-step Fourier Method at time $t$, then for any $\epsilon > 0$, there are a positive integer $n$, $m$ points $x_1, ..., x_m \in K$, and real constants $c_i$, $\theta_i$, $\xi_{ij}$, $i = 1, ..., n$, $j = 1, ..., m$, such that for*

$$R_\phi := \sum_{i=1}^{n} c_i \sigma \left( \sum_{j=1}^{m} \xi_{ij} u(x_j) + \theta_i \right), \tag{8}$$

$$\left| \Psi_{t+\Delta t}(\Psi_t)(x) - \mathcal{F}^{-1}(R_\phi \cdot \mathcal{F}(v_t))(x)) \right| < \epsilon \tag{9}$$

*holds for all $u \in V$.*

*Proof.* See Appendix B.

## 3.2  Data Generation

The training and testing datasets are constructed based on varying pump profiles, $P(\boldsymbol{r})$, as elucidated in Equation (4). This profile is characterized by four Gaussian spots, represented by $N_G = 4$. Out of these, three spots have their power adjusted to exceed the threshold, specifically $p_i = 1.6\,p^{\text{th}}$, while the power of the remaining spot is set below the threshold at $p_i = 0.5\,p^{\text{th}}$. Each spot's spatial profile is represented as $G_i(\boldsymbol{r})$. Moreover, the boundary conditions that the condensate density vanishes near the computation frame in both real and Fourier space are also applied.

These profiles are stochastically determined within a square region that measures $64\,\mu\text{m} \times 64\,\mu\text{m}$ out of the entire configuration with $128\,\mu\text{m} \times 128\,\mu\text{m}$. The region where Gaussian stops stay is smaller than the full grid, to make sure that they are still far from the region where the boundary condition is applied. Care has also been taken to ensure that the Gaussian spots are non-overlapping. Additionally, every pump profile is unique, and among the spots with power exceeding the threshold, each one is distinct from the others, thereby eliminating any potential redundancy. Given $0.5\,\mu\text{m}$ resolution per pixel per dimension, the total datasets for pump configuration is with size $256 \times 256 \times 1568$ where 256 represents each square map size per dimension and 1568 is the number of different pump configurations (of which 314 configurations are used for testing and 1254 for training respectively). The datasets for the density map is with size $256 \times 256 \times 2 \times 1568$ where 2 refers to the density at the initial and final time.

It is worth mentioning that the systems of all the datasets are chosen with system only at stationary state with single energy mode, which means that the results with multiple energy modes are excluded. In multimode cases, the wavefunction density changes at different times, which can be found in experiments [56, 57].

For better emulating the experiment $\sigma \approx 0.85\,\mu\text{m}$, standing for FWHM of each Gaussian spot equaling to $2\,\mu\text{m}$, is chosen. The simulation is based on InGaAs QWs [58] with slightly negatively detuned cavities. The parameters are the following: $m = 0.28\,\text{meV}\,\text{ps}^2\,\mu\text{m}^{-2}$, $|\chi|^2 = 0.4$, $N_{\text{QW}} = 6$, $g_0 = 0.01\,\text{meV}\,\mu\text{m}^2$, $\hbar R = 10g$, $\eta = 2$, and $\gamma^{-1} = \Gamma^{-1} = 5.5\,\text{ps}$.

## 4  Results

In Figure 4, we present model predictions for 4 representative test cases. Note that we have carefully chosen four distinct pump profiles to visualize model performance over varying inputs. As we see from the second row of the figure, the model is highly accurate in predicting the steady-state solution for the GPE. It is not only able to predict accurate values at pump locations and their locality but it is also able to predict direction of flow for the wave fronts as well as the scattering from the barrier produced from the spot below the threshold. Where the model exceeds expectations most is in the detail to which it also captures the interference pattern for different pump configurations. We see an almost identical pattern, including the parity of fringes among spots, between the predictions and simulation ground truths. The parity of these fringes are responsive to the distance between spots (see experiment in [57]), which also indicates that our model is capable of capturing these details.

Looking at error plots between predictions and ground truth values, we see that the highest errors occur very close to the pump locations. This is due to the high degree of nonlinearity which is exhibited near this region and the information loss that occurs from the Fast Fourier Transform cutoff modes in the FNO architecture. We see other smaller errors around the edges of wave fronts which could potentially be tackled using physics-informed gradient losses, which we intend to test in future work.

We present a full overview of cell averaged absolute difference errors in Figure D1 in Appendices D for all 314 test cases. The 4 test predictions shown in Figure 4 correspond to cases 75, 97, 181 and 291 respectively. The cell averaged errors on the test set range between $1.774 \times 10^{-2}$, for case 198, and $8.080 \times 10^{-2}$, for case 31. Empirically, the lower errors correspond to pump configurations where distance between the pumps is smaller, leading to better interference predictions. The higher errors correspond to pump configurations where at least one pump is far away from the other two,

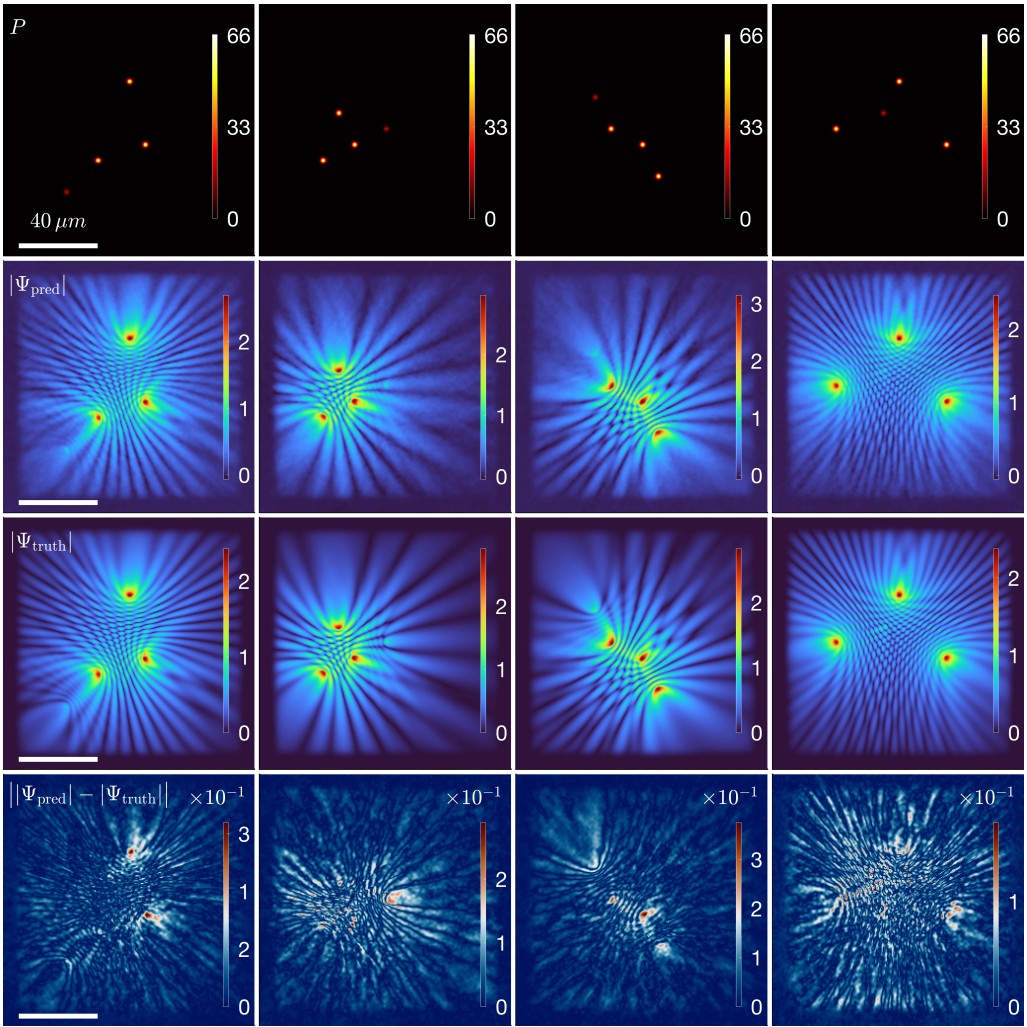

Figure 4: Comparison of the prediction from the Fourier Neural Operator approach. The columns, left to right, show different pump configurations. The rows, from top to bottom, are pump profiles $P$, predictions $|\Psi_{\text{pred}}|$, ground truths $|\Psi_{\text{truth}}|$, and errors $\left||\Psi_{\text{pred}}| - |\Psi_{\text{truth}}|\right|$.

leading to worse interference pattern predictions. This is evident in the results from Figure D2 in Appendices D where pumps are very far apart as opposed to Figure 4 where pumps are closer together.

We found that, when we trained the FNO with a cutoff smaller than 128 Fourier modes for both spatial directions, which is conventional, that the model was not able to capture nonlinearities in the system to a great extent and that it would take longer to converge. We also observed that predictions were accurate in a large square sub-region of the grid but performance dropped outside this region. This was likely due to the fact that with a $256 \times 256$ grid, in order to capture the finer details, we would need to also include higher frequency modes of the Fourier expansion. To this avail, we increased the cutoff to 128 Fourier modes in both directions and observed a significant improvement in performance. For larger grid sizes, there is a higher probability that the distance between the randomly generated pump locations will be greater. This would require us to increase further the number of Fourier modes to retain similar levels of accuracy or employ dimensionality reduction techniques.

The hyperparameters are detailed in Appendix C. It is worth noting that the usage of the domain padding can reduce the antifacts around the boundary conditions. Moreove, the loss function named 'H1Loss' (see Appendix C) considers not only its true values but also in their derivatives. With both the FNO model and numerical solver run in parallel computing using a GPU, the time to predict

solutions for 314 cases using the CUDA-based numerical method took 2808.03 s, whereas the FNO model took 2.66 s.

# 5 Future Work

In this work, we use numerical datasets for training and prediction. However, in the future we hope to directly use experimentally collected data to train the model on ground truth physics. Pump profiles and wavefunction density maps shown in Figure 4 can be obtained directly from spatial light modulator and photoluminescence spectroscopy, respectively. Due to the improvement in semiconductor fabrication, clear interference patterns, which are very close to those simulated shown here, can be found from inorganic semiconductor materials [57, 59]. Furthermore, with the help of a streak camera, photoluminescence can be captured at the picosecond level, which makes it possible to make predictions of a time-resolved condensate formation on the basis of purely experimental data.

Various general ML methods have been proposed to incorporate underlying physics-based losses and information into the model to aid the learning task, such as in [60–63]. In this work, we have taken a purely data-driven approach to training, however, we believe that incorporating additional physics-informed loss terms will strictly increase the rate of convergence and accuracy. This is especially appealing given that we have a strong theoretical understanding of the underlying system.

In the future, we aim to also propose a novel Neural Operator architecture entirely, tailored to closer align to the computational procedure of the SSFM. As shown in Appendices A and B, the FNO can be seen as a first-order approximation to the SSFM. We aim to instead to take structure from the BCH formula at second-order and embed this in the model. This will better guide convergence dynamics in the weight landscape, which in theory will produce more accurate solutions at a faster rate.

# 6 Conclusions

In the present study, we explored the potential of the FNO in the context of polariton condensates. Our findings, as detailed in Section 4, demonstrate a notable alignment with the simulation data with an approximate $1000\times$ speed up in solution generation compared to CUDA-based GPU solvers. This research paves the way for the conceptualization and development of advanced large-scale all-optical devices. Furthermore, this method draws parallels with the principles of EDA traditionally used in chip design. It introduces an innovative avenue to meet the growing demand for fast and reliable solutions in the realm of all-optical chip design.

# 7 Acknowledgements

We are greatly indebted to Prof. Pavlos Lagoudakis for giving feedback on this manuscript.

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

## Appendix A    Split-step Fourier method

In this Appendix, the numerical solution to Equations (1) and (2) is derived using SSFM, which serves as the supplementary information for Section 3.1. Let us start with the GPE first and Equation 1 can rearranged as

$$\frac{\partial}{\partial t}\Psi = \left\{i\frac{\hbar}{2m}\nabla^2 - i\frac{\alpha}{\hbar}|\Psi|^2 - i\frac{G}{\hbar}\left[\mathcal{N} + \frac{\eta}{\Gamma}P\right] + \frac{1}{2}\left[R\mathcal{N} - \gamma\right]\right\}\Psi. \tag{A1}$$

The direct solution from Equation (A1) at time interval $[t, t + \Delta t]$ is

$$\Psi_{t+\Delta t} = \exp\left[(\widehat{f_L} + \widehat{f_N})\Delta t\right]\Psi_t, \tag{A2}$$

where

$$\widehat{f_L} = i\frac{\hbar}{2m}\nabla^2, \tag{A3}$$

$$\widehat{f_N} = -i\frac{\alpha}{\hbar}|\Psi|^2 - i\frac{G}{\hbar}\left[\mathcal{N} + \frac{\eta}{\Gamma}P\right] + \frac{1}{2}\left[R\mathcal{N} - \gamma\right]. \tag{A4}$$

Note that since $[\widehat{f_L}, \widehat{f_N}] \neq 0$, the exponentiation identity cannot be applied directly, namely, $\exp[\widehat{f_L} + \widehat{f_N}] \neq \exp[\widehat{f_L}]\exp[\widehat{f_N}]$. By applying the Baker–Campbell–Hausdorff (BCH) formula at second order and the strang splitting, we have

$$\Psi_{t+\Delta t} = \exp\left[\frac{1}{2}\widehat{f_N}\Delta t\right]\exp\left[\widehat{f_L}\Delta t\right]\exp\left[\frac{1}{2}\widehat{f_N}\Delta t\right]\Psi_t, \tag{A5}$$

which give the accuracy of $\Delta t^3$. The essence of SSFM is that it can convert the nonlinear operator $\widehat{f_N}$ into the linear one through the Fourier transform. We can always construct the relation that

$$\frac{\partial}{\partial t}\Psi = \widehat{f_L}\Psi. \tag{A6}$$

Then, applying the Fourier transform for above equation, we have

$$\frac{\partial}{\partial t}\mathcal{F}[\Psi] = \widehat{f_P}\mathcal{F}[\Psi], \tag{A7}$$

where

$$\widehat{f_P} = -i\frac{\hbar}{2m}|\boldsymbol{k}|^2, \tag{A8}$$

is the momentum operator. This implies that Equation A6 can be rewritten as

$$\Psi_{t+\Delta t} = \exp\left[\frac{1}{2}\widehat{f_N}\Delta t\right]\mathcal{F}^{-1}\left[\exp\left[\widehat{f_P}\Delta t\right]\mathcal{F}\left[\exp\left[\frac{1}{2}\widehat{f_N}\Delta t\right]\Psi_t\right]\right]. \tag{A9}$$

Ultimately, the final state wavefunction can be obtained by iteraction of Equation (A9) overall the infinitesimal time steps. During each iteration, $\mathcal{N}$ is updated within the operator $\widehat{f_N}$. That is

$$\mathcal{N}_{t+\Delta t} = \exp\left[(-\Gamma + R|\Psi|^2)\Delta t\right]\mathcal{N}_t + P\Delta t. \tag{A10}$$

The kernel defined by Equation (A5) is used to calculate the ground truth for the simulation with BCH at second order. However, if we limit ourselves at BCH at first order, then corresponding Equation (A9) can be simplified as

$$\Psi_{t+\Delta t} = \mathcal{F}^{-1}\left[\exp\left[\widehat{f_P}\Delta t\right]\mathcal{F}\left[\exp\left[\widehat{f_N}\Delta t\right]\Psi_t\right]\right]. \tag{A11}$$

From here, Equation (A11) takes the functional form of the Kernel Integral Operator of the FNO introduced in Section 3.1. This is stated formally in the SSFM - FNO correspondence given in Theorem 1.

## Appendix B    Theorems

**Theorem B1.** *(Universal Approximation Theorem for Functionals.)* *Suppose that $\sigma \in (TW)$ is a Tauber-Wiener function, $X$ is a Banach Space, $K \subset X$ is a compact set, $V$ is a compact set in $C(K)$, $f$ is a continuous functional defined on $V$, then for any $\epsilon > 0$, there are a positive integer $n$, $m$ points $x_1, ..., x_m \in K$, and real constants $c_i$, $\theta_i$, $\xi_{ij}$, $i = 1, ..., n$, $j = 1, ..., m$, such that*

$$\left| f(u) - \sum_{i=1}^{n} c_i \sigma \left( \sum_{j=1}^{m} \xi_{ij} u(x_j) + \theta_i \right) \right| < \epsilon \tag{B1}$$

*holds for all $u \in V$.*

*Proof.* See [S34] and [S64–S66].

**Theorem B2.** *(Universal Approximation Theorem for Operators.)* *Suppose that $\sigma \in (TW)$ is a Tauber-Wiener function, $X$ is a Banach Space, $K_1 \subset X$, $K_2 \subset \mathbb{R}^d$ are two compact sets in $X$ and $\mathbb{R}^d$, respectively, $V$ is a compact set in $C(K_1)$, $G$ is a nonlinear continuous operator, which maps $V$ into $C(K_2)$, then for any $\epsilon > 0$, there are positive integers $n$, $p$, $m$, constants $c_i^k$, $\xi_{ij}^k$, $\theta_i^k$, $\zeta_k \in \mathbb{R}$, $w_k \in \mathbb{R}^d$, $x_j \in K_1$, $i = 1, ..., n$, $k = 1, ..., p$, $j = 1, ..., m$, such that*

$$\left| G(u)(y) - \sum_{k=1}^{p} \sum_{i=1}^{n} c_i^k \sigma \left( \sum_{j=1}^{m} \xi_{ij}^k u(x_j) + \theta_i^k \right) \sigma(w_k \cdot y + \zeta_k) \right| < \epsilon \tag{B2}$$

*holds for all $u \in V$ and $y \in K_2$.*

*Proof.* See [S34] and [S64, S67, S66].

**Theorem B3.** *SSFM-FNO Correspondence.* *Suppose that $\sigma \in (TW)$ is a Tauber-Wiener function, $X$ is a Banach Space, $K \subset X$ is a compact set, $V$ is a compact set in $C(K)$, $\Psi_t$ is a nonlinear continuous operator representing the solution of the first-order Split-step Fourier Method at time $t$, then for any $\epsilon > 0$, there are a positive integer $n$, $m$ points $x_1, ..., x_m \in K$, and real constants $c_i$, $\theta_i$, $\xi_{ij}$, $i = 1, ..., n$, $j = 1, ..., m$, such that for*

$$R_\phi := \sum_{i=1}^{n} c_i \sigma \left( \sum_{j=1}^{m} \xi_{ij} u(x_j) + \theta_i \right), \tag{B3}$$

$$\left| \Psi_{t+\Delta t}(\Psi_t)(x) - \mathcal{F}^{-1}(R_\phi \cdot \mathcal{F}(v_t))(x) \right| < \epsilon \tag{B4}$$

*holds for all $u \in V$.*

*Proof.* Limiting the solution of Equation 1 to first-order in the BCH formula, we get

$$\Psi_{t+\Delta t}(\Psi_t)(x) = \mathcal{F}^{-1} \left[ \exp\left[\widehat{f}_P \Delta t\right] \mathcal{F} \left[ \exp\left[\widehat{f}_N \Delta t\right] \Psi_t \right] \right]. \tag{B5}$$

Also considering the Kernel Integral Operator for the FNO as defined in Equation 7, we have

$$(\mathcal{K}_{t+1}(v_t))(x) = \mathcal{F}^{-1}(R_\phi \cdot \mathcal{F}(v_t))(x) \qquad \forall x \in \mathbb{R}^n \tag{B6}$$

It is clear that these two equations are equivalent for some $R_\phi = \exp\left[\widehat{f}_P \Delta t\right]$ and $v_t = \exp\left[\widehat{f}_N \Delta t\right]\Psi_t$.

Now, appealing to the Universal Approximation Theorems B1 and B2, and using the fact that a non-linearity $\sigma$ is applied following every Kernel Integral layer, it is clear that there exists some

learnable $R_\phi$ which can approximate $\exp\left[\widehat{f}_P \Delta t\right]$, and indirectly learnable $v$ which can approximate $\exp\left[\widehat{f}_N \Delta t\right]\Psi_t$, to an arbitrary margin $\epsilon$. It directly follows that the Fourier Kernel Integral Operator can approximate the first-order SSFM time step update to an arbitrary degree of accuracy which then implies that the FNO as a whole can approximate the solution to the GPE to an arbitrary degree of accuracy, with the number of Kernel Integral layers in the FNO dictating the time discretization $\Delta t$ of the approximation.

## Appendix C    Model Hyperparameters

Table C1: Model Hyperparameters

| Hyperparameter | Value |
| --- | --- |
| Learning Rate | 0.001 |
| Batch Size | 32 |
| Epochs | 560 |
| Optimizer | Adam |
| Loss Function | H1Loss |
| Fourier Layers | 4 |
| Height Modes | 128 |
| Width Modes | 128 |
| Hidden Dimension | 64 |
| Domain Padding | 0.125 |
| Domain Padding Mode | symmetric |
| Activation Function | GeLU |

The training loss as a function of the epochs is shown in Figure C1. The model was trained on an NVIDIA RTX 4090 GPU with an Intel i9 13900KF and each epoch took on average 15.1s to train.

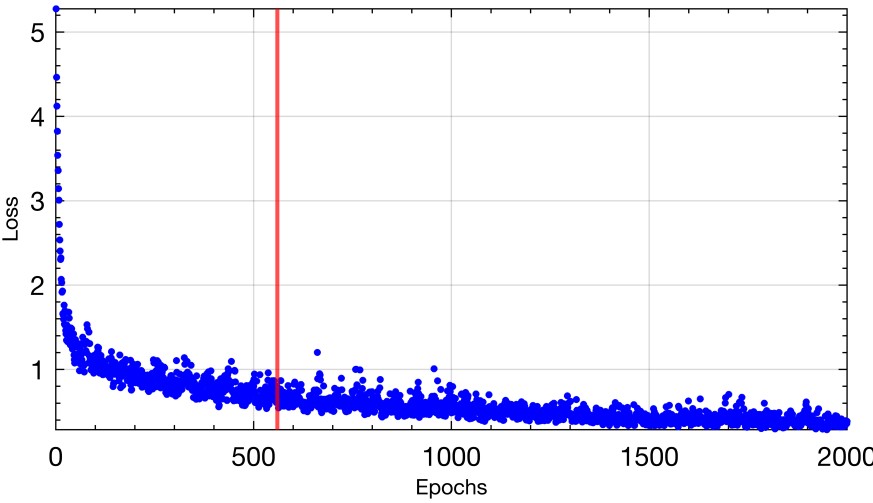

Figure C1:  Training Loss per Epoch for Fourier Neural Operator. The red line, representing epoch number $560$, is selected for the purpose of prediction analysis.

## Appendix D    Extended Results

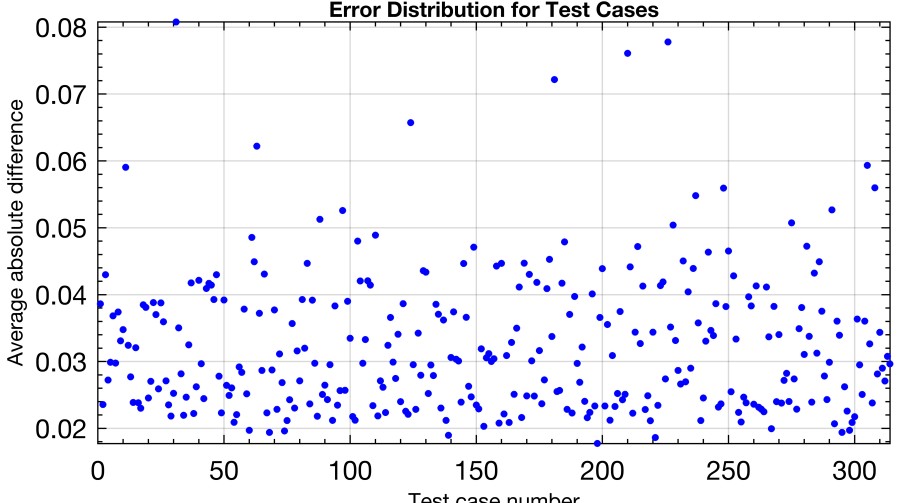

Figure D1: Average absolute difference for each test case normalised by grid size.

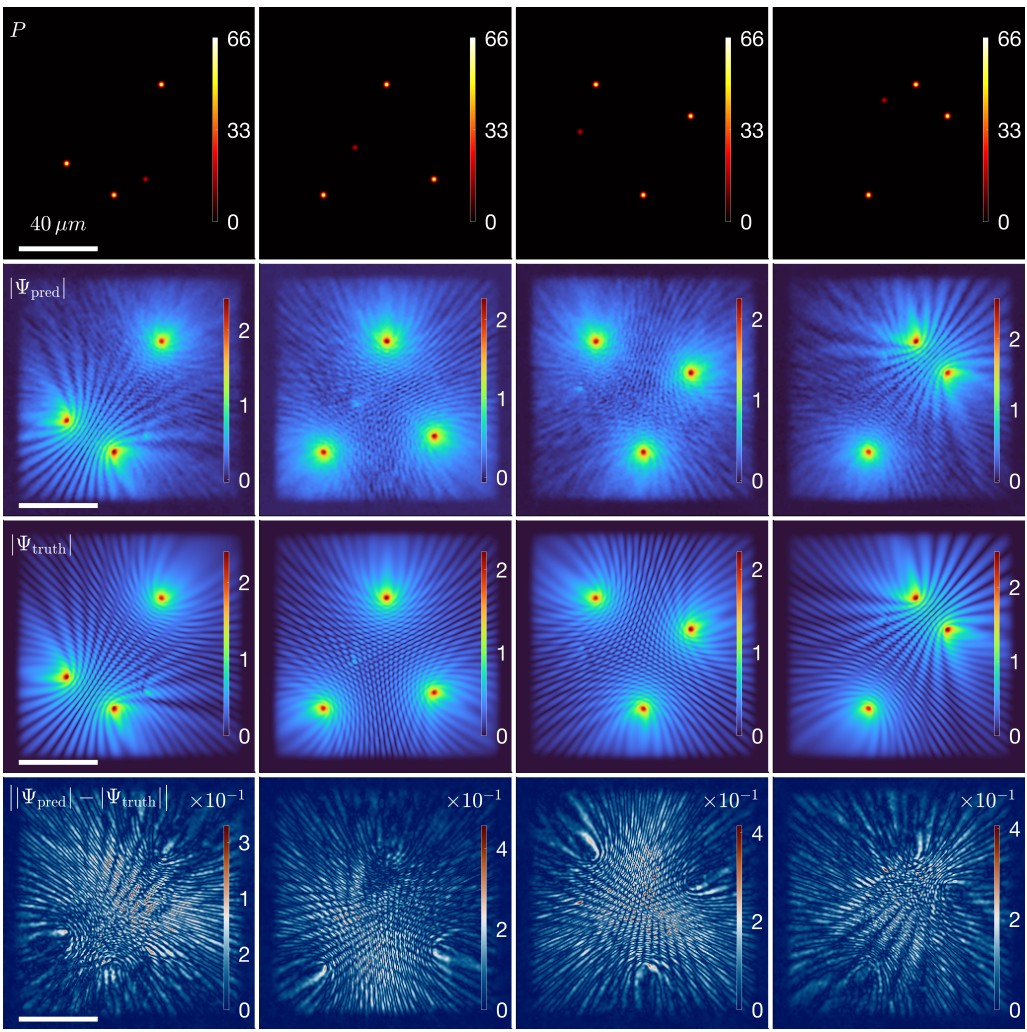

Figure D2: Comparison of the prediction from the Fourier Neural Operator approach. The columns, left to right, show different pump configurations(test case number 69, 103, 148, and 258 from Figure D1). The rows, from top to bottom, are pump profiles $P$, predictions $|\Psi_{\text{pred}}|$, ground truths $|\Psi_{\text{truth}}|$, and errors $\big||\Psi_{\text{pred}}| - |\Psi_{\text{truth}}|\big|$.

