# OpenReview forum: "Exciton-Polariton Condensates: A Fourier Neural Operator Approach"
_NeurIPS.cc/2023/Workshop/AI4Science — NeurIPS2023-AI4Science Poster_

### Official Review · Reviewer_r45m · 2023-10-07
**Ok paper applying FNO to solve Gross-Pitaevskii equations**

**Rating:** 6
**Confidence:** 4

**Review:**

The authors proposed to use FNO for solving Gross-Pitaevskii equations. It is very well written, with nice figures and good experiments. However the novelty is lacking here and no clear or convincing points why FNO is better intuitively is provided.

---

### Official Review · Reviewer_qiTs · 2023-10-19
**Neural operator for polariton wave functions**

**Rating:** 8
**Confidence:** 2

**Review:**

This paper presents the use of Fourier neural operators to solve the Gross-Pitaevskii equation describing polariton condensates. The paper contains an extensive introduction to the topic, and explains the standard solving method (SSFM) before describing in details the neural operator. The training setup is well described and the paper seems to be reproducible. Thei achieve a good accuracy with much faster computing time. For these reasons, this is an interesting paper and the method proposed seems promising.

---

### Meta-Review · Area_Chair_5K6X · 2023-10-26

**Recommendation:** Accept (Poster)
**Confidence:** 4

**Metareview:**

The author deal with The paper addresses the challenge of simulating complex nonlinear systems in exciton-polariton condensates used in all-optical devices. They propose the use of a machine-learning-based Fourier Neural Operator approach, which can predict final-state solutions with high accuracy and much faster than traditional GPU solvers, potentially facilitating the development of all-optical chip design workflows.

Even though reviewer r45m thinks the paper lacks the intuition as to why FNO is better, this paper is still a good practice of machine learning application.